# Correlative Monitoring of Immune Activation and Tissue Damage in Malignant Melanoma—An Algorithm for Identification of Tolerance Breakage During Immune Checkpoint Inhibitor Therapy of Cancer

**DOI:** 10.3390/ijms21062020

**Published:** 2020-03-16

**Authors:** Renate U. Wahl, Marike Leijs, Arturo Araujo, Albert Rübben

**Affiliations:** 1Department of Dermatology, Euregio Skin Cancer Center, RWTH Aachen University, Pauwelsstraße 30, 52074 Aachen, Germany; rwahl@ukaachen.de (R.U.W.);; 2The AIWorks Research Lab, 7 Gower St, London WC1E 6HA, UK; arturo@cancerevo.org; 3Department of Computer Science, University College London, Malet Place, London WC1E 6BT, UK; 4Department of Dermatology, Sint-Nikolaus Hospital, Hufengasse 4-8, 4700 Eupen, Belgium

**Keywords:** treatment monitoring, tolerance breakage, checkpoint inhibitor therapy, temporal correlation, malignant melanoma, interleukin 6, S100B, eosinophils, macrophages, irAE

## Abstract

We describe an innovative approach for identification of tolerance breakage during immune checkpoint inhibitor therapy in malignant melanoma. Checkpoint inhibitor therapy enhances the immunologic clearance of cancer by suppressing pathways which induce immune suppression and tolerance. We posit that by analyzing temporal correlations of key markers of immune activation and tissue damage it would be possible to detect the onset of anticancer immune reaction as well as of immunologic adverse effects which might become crucial for optimization as well as safety of immune checkpoint inhibitor treatment. We analyzed time courses of routine laboratory values of serum tumor markers as well as of markers of immune activation in 17 patients with metastasized malignant melanoma receiving checkpoint inhibition and weekly laboratory controls. A parallel serum level increase of interleukin-6 and the tumor marker S100B could be identified in 13 patients, suggesting that the onset of tolerance breakage under checkpoint inhibition may be identified and measured. Immune-related adverse events in the patients were also accompanied by a peak of IL-6. In six patients, the onset of a putative anticancer immune reaction and the beginning of immunologic adverse events occurred in the same treatment cycle; in six patients the immunologic adverse reactions took place in separate cycles.

## 1. Introduction

Checkpoint inhibitor therapy has been a major breakthrough in anticancer therapy leading to durable responses in many cancer entities such as malignant melanoma, cutaneous squamous cell carcinoma, Merkel cell carcinoma, lung cancer, classical Hodgkin lymphoma, squamous cell cancer of the head and neck, renal cancer, bladder cancer, endometrial carcinoma, breast cancer, liver cancer, gastric cancer and colorectal cancer [1,2]. Treatment is performed by administration of monoclonal antibodies which block the coinhibitory surface receptors or their ligands: programmed cell death 1 (PD-1) and cytotoxic T-lymphocyte antigen 4 (CTLA4) on immune cells as well as programmed cell death ligand 1 (PD-L1) on immune cells and tumor cells. The blockade of immunosuppressive pathways results in an enhanced immune response against tumor antigens [1,3]. Nevertheless, not all patients respond to checkpoint inhibitors and the treatment itself is associated with a high risk of severe side effects due to therapy induced autoimmune inflammation in various tissues. Especially combined checkpoint inhibition targeting CTLA4 as well as PD-1 is associated not only with the best therapeutic response but also with a significantly increased rate of severe immune-related adverse events (irAEs) [4]. Hence, research has focused on predictive biomarkers in order to determine prior to therapy which patients and tumors might respond to checkpoint inhibitors and which patients might develop severe side effects [5,6,7,8,9,10,11,12,13,14,15,16,17,18].

Table 1 provides an overview of predictive biomarkers in the strict sense that these biomarkers have been determined prior to the start of therapy.

An alternative approach to the search of predictive biomarkers sensu stricto is the identification of predictive biomarkers which are determined under checkpoint inhibitor therapy but before evidence by radiologic imaging of response to therapy [18,21,22]. Table 2 provides an overview of treatment response monitoring biomarkers. Although this class of predictive biomarkers is not useful for a preselection of patients receiving checkpoint inhibitor anticancer therapy, these biomarkers might nevertheless help identify patients at an earlier time point who might benefit from therapy continuation or therapy escalation, from dose reduction or interval prolongation of checkpoint inhibitor therapy, or from an early switch to an alternative treatment. It has to be noted that the occurrence of immune-related adverse events under checkpoint inhibition is, by itself, an indicator of a better anticancer response in malignant melanoma [23]. Still, evidence is lacking as to whether checkpoint inhibitor treatment has to be continued until the tolerable maximum of immune-related adverse events has been reached or if the patient might profit from an earlier discontinuation or dose reduction considering that the treatment of severe immune-related adverse events consists of immunosuppression which in turn could jeopardize the anticancer efficacy of checkpoint inhibition.

We have developed this alternative approach further by searching for biomarkers in the blood of patients which might allow for determining the point in time of tolerance breakage during checkpoint-inhibitor therapy of cancer.

In analogy to autoimmune diseases, the presented approach relies on the premise that immune-mediated tissue damage, as measured by the increase of proteins released in the blood from damaged tissues, is a surrogate marker of immunologic activity within the tissue. Likewise, anticancer immune responses should induce a release of cancer-derived proteins or DNA into the blood of the treated patients due to destruction of the tumor.

Therapy induced tolerance breakage during checkpoint-inhibitor treatment might therefore be assumed when an increase of DNA or protein release from cancer tissue coincides chronologically with a measurable increase of biomarkers in the blood or serum which are associated with an immune activation (Figure 1, timepoint t2). Likewise, tolerance breakage would also occur when the immune activation coincides with the release of proteins from damaged tissue due to treatment-induced autoimmunity (Figure 1, timepoint t3). A prerequisite for this assumed correlation is a non-continuous phasic increase of the immune reaction (Figure 1, graph I2) at the time of immune-mediated tissue damage as it has been postulated for interferon-γ (IFNγ) release and T-cell activation [24].

Therapy monitoring of human autoimmune diseases has identified several measurable blood and serum biomarkers which correlate with immune activity (Table 3). In autoimmunity, the used biomarkers vary depending on the kind of autoimmune disease reflecting different immunologic response mechanisms [25,26,27,28,29]. Therefore, in the presented study a panel of different biomarkers suggesting immune activation has been screened retrospectively for a temporal association with either the release of cancer-derived proteins in the serum or the rise of proteins from tissues which have been damaged by checkpoint inhibitor-induced inflammation. This approach has been tested for melanoma checkpoint inhibitor treatment in order to provide proof of principle, but it could represent a universal algorithm which might also be applicable to other cancer entities.

The next paragraph deals with theoretical considerations on the choice of biomarkers for detection of immune activation as well as for monitoring tissue damage of cancer cells and of organs targeted by immune-mediated side effects.

## 2. Theoretical Considerations

Besides the requisite of choosing biomarkers for monitoring the immune activation which display a non-continuous and phasic activation time course, the choice of biomarkers and the optimal time point for sampling is expected to be determined mainly by the presumed temporal dynamics of the immune reaction. In experimental settings, immune cells release cytokines within hours and subsequent activation of the respective immune function seems to last only several days [24]. Little is known on the dynamics of immune activation in human patients, but levels of cytokines or acute phase proteins seem to oscillate in the range of one day to one month, depending on the analyzed biomarker [30]. In addition, the dynamics of release of cancer derived proteins and DNA have to be considered as well. Cancers constantly release proteins and DNA due to continuous cell death associated with tumor cell turnover. For malignant melanoma it has been estimated that the cell loss fraction is up to 70% [31]. The release of cancer-associated proteins therefore depends on the amount of proteins in the cancer cells, on the amount of secreted proteins, on the total tumor mass, on the specific cell loss fraction of the tumor and on cancer cell cycle and cancer growth fraction (Figure 2) [32]. An immune-mediated loss of tumor mass should result in a temporary increase in the release of proteins or DNA in the serum followed by a reduction in biomarker release, which in the case of successful treatment should be lower than the initial value due to reduced total cancer mass (Figure 1, C2).

## 3. Results

Time courses of routine laboratory values of 17 patients with metastasized malignant melanoma receiving checkpoint inhibitor treatment and weekly laboratory controls were reviewed for a temporal correlation between potential biomarkers of immune activation and routine laboratory values used for detection of immune-related adverse events as well as release of tumor markers. Blood cell count with white cell differentiation, c-reactive protein (CRP) and interleukin-6 (IL-6) in the serum were examined as potential biomarkers of immune activation due to their association with autoimmune and inflammatory diseases (Table 3). S100B and lactate dehydrogenase (LDH) served as serum tumor markers for monitoring malignant melanoma [33,34]. Five patients received monotherapy with the anti-CTLA4 antibody ipilimumab, 13 patients were treated with either nivolumab or pembrolizumab which are anti-PD-1 antibodies and six patients were treated with a combination of ipilimumab with nivolumab (Table 4).

The first patient treated with ipilimumab monotherapy had the longest follow-up dating from the time of introduction of ipilimumab for malignant melanoma. Besides CRP, IL-6 was closely monitored due to the association with inflammatory bowel disease [25]. Figure 3 displays the time course of CRP, IL-6, LDH and S100B during treatment. All values were normalized with 100% indicating the upper limit of normal (ULN). LDH did not change significantly during treatment but S100B demonstrated a peak on day 62 after starting immunotherapy which coincided with an increase of CRP and IL-6. In direct comparison, IL-6, although at levels below ULN, seemed to react earlier and with a greater relative amplitude to ipilimumab administration than CRP and the impression was that IL-6 increased with every ipilimumab infusion, which suggests a treatment-induced immunologic booster effect resulting in immune oscillations as previously postulated for CRP in cancer patients [35]. A peak of S100B which coincided with an IL-6 increase could be observed in three other patients treated with ipilimumab monotherapy occurring between the first day after starting ipilimumab (=day 1) and day 81 (Table 4). This observation suggests that treatment and immune-induced destruction of cancer tissue which should represent tolerance breakage by checkpoint inhibition might be visualized by a close monitoring of IL-6 and S100 as postulated and described by Figure 1. In some patients, a slight increase of the tumor marker LDH could be observed as well but changes in S100B were significantly more distinct. Two of the five patients treated with ipilimumab monotherapy were long-term responders and were treated subsequently with nivolumab or pembrolizumab (Table 4).

Figure 4 shows the time course of IL-6, S100B and creatine kinase (CK) during treatment of patient no. 5 with pembrolizumab which resulted in complete remission of the melanoma metastases. A strong S100B tumor marker release in response to pembrolizumab treatment with concomitant IL-6 release was observed at day 8. A second smaller oscillation of both S100B and IL-6 occurred in the following cycle. S100B continued to decrease below pretreatment levels until the end of cycle 6. Using the tumor marker model depicted by Figure 2, this decrease of S100B should reflect a slow reduction of total tumor mass as hypothesized under efficient checkpoint inhibitor therapy by Figure 1, graph C2. Increase of IL-6 in cycle 7 was associated with both an increase in S100B suggesting a second immunologic reaction against melanoma metastases as well as an increase in creatine kinase (CK) which corresponded to immune-mediated myositis in the patient and which had to be treated with corticosteroids.

Patient no. 6 did not respond to pembrolizumab treatment. Figure 5 denotes the temporal behavior of S100B and IL-6 in this patient. Administration of checkpoint inhibitor resulted in a steep and parallel decline of both S100B and IL-6, especially in the first cycle. This unexpected behavior was observed in patient 4 under nivolumab treatment as well and was termed a “paradox” reaction in Table 4. Using the tumor marker model depicted by Figure 2, this decrease of S100B might be explained by a yet not understood effect of PD-1 blockade on melanoma cell cycle resulting in partial cell cycle arrest and reduced S100B release. It has been observed that melanoma cells may rely on an autocrine secretion of IL-6 [36]. The parallel decrease of IL-6 might indicate that PD-1 blockade interferes with this autocrine IL-6 loop. The increase of S100B in treatment cycle 5 corresponded clinically to a massively progressive disease. Without this clinical information, rise of S100B and IL-6 would have been interpreted as immune activation against melanoma metastases.

Combined immune checkpoint inhibitor therapy with nivolumab and ipilimumab was performed in the patients either as first checkpoint inhibitor treatment or after a monotherapy which did not result in sufficient clearance of the tumor. Patient no. 8 only demonstrated partial remission after 18 cycles of nivolumab and was therefore switched to combination therapy. Figure 6 displays the last two cycles of monotherapy and the first two cycles of combination therapy. Tolerance breakage was observed after the second administration of combination therapy. A strong increase in IL-6 and CRP was paralleled by a strong rise of S100B as well as of liver enzymes alanine aminotransferase (ALT) and aspartate transaminase. Immune hepatitis in this patient necessitated prolonged oral glucocorticoid treatment with prednisolone, still the patient demonstrated complete response of the melanoma metastases and long-term remission. After two cycles of combination therapy, the treatment was continued with a further eight cycles of nivolumab monotherapy (Table 4). Oscillations of CRP as well as IL-6 form a double peak before intervention with corticosteroids (Figure 6).

Figure 7 shows the time courses in two other melanoma patients who responded with complete regression of their metastases. Patient no. 15 had already responded to combination immunotherapy with nivolumab and ipilimumab but developed a relapse one year later. Nivolumab monotherapy was started and myositis with an increase of myoglobulin was observed at day 98 with a parallel increase of IL6 as well as of S100B (Figure 7A).

Figure 7B shows immune activation and parallel tumor marker release as well as the effect of corticosteroid treatment in patient no. 16 after only one administration of combined immunotherapy with nivolumab and ipilimumab. The patient first developed a skin rash and then hypohysitis and pneumonitis requiring corticosteroid intervention. Interestingly, tumor marker release in this patient continued despite suppression of IL-6 by prolonged corticosteroid treatment.

Figure 8 demonstrates immune activation and parallel tumor marker release in a melanoma patient with high tumor burden under combination therapy with nivolumab and ipilimumab. Treatment led to an increase of S100B which persisted during two cycles followed by a steep decreased to normal values. As observed in other patients, LDH levels were not informative. The patient experienced near total reduction of metastases. After four cycles of nivolumab and ipilimumab, the treatment was continued with nivolumab monotherapy.

In six patients, the onset of a putative anticancer immune reaction and the beginning of immunologic adverse events occurred in the same treatment cycle; in six patients the immunologic reactions took place in separate cycles. Nevertheless, temporal separation could be observed in some patients even within one cycle (Table 4, Figure 5).

## 4. Discussion

The presented biomarker study analyzed the time courses of routine laboratory values and of serum tumor markers as well as of markers of immune activation in 17 patients with metastasized malignant melanoma receiving immune checkpoint inhibitor therapy and weekly laboratory controls. In 13 patients we could observe a parallel increase of tumor marker S100B and of the immune activation marker IL-6 in the serum. We hypothesize that this molecular signature indicates tolerance breakage due to immune checkpoint inhibition leading to an enhanced anticancer activity.

The presented retrospective analysis may not provide definite proof of actual tolerance breakage taking place at the supposed time point within the metastases. Definite proof would have required for each patient sequential biopsies taken from one melanoma metastasis without removing the metastasis at multiple time points during therapy, an approach which would have been technically difficult, and which would have encountered severe ethical concerns. An alternative approach for future studies could be sequential PET-imaging under therapy using the enhanced uptake of appropriate marker molecules in the metastases as a surrogate for local immune activation [37]. Nevertheless, the observation of similar serum kinetics of IL-6 during putative anticancer immune reaction as well as at the time of severe immune-related adverse events in the analyzed patients suggests that concomitant IL-6 and S100B release might indeed reflect a rise of anticancer immune activity and tolerance breakage.

Besides providing first evidence that tolerance breakage within the cancer tissue under immune checkpoint inhibition treatment of human cancers might be identified, monitored and measured, the presented study also contributes insight into the temporal dynamics of immune regulation under checkpoint inhibition. In the analyzed patients, rise of cytokine IL-6 as well as release of tumor marker S100B demonstrated oscillations with a frequency in the range of a few days, suggesting that weekly monitoring of these markers constitutes a minimal requirement in order to identify the expected effects. In future studies involving melanoma or other cancers, this time frame should be considered. A recent publication analyzed CRP every day to every second day in a case of checkpoint inhibitor associated neutropenia and the published time course of CRP in Figure 1 of this publication resembles our time courses of CRP in Figure 3 and Figure 6 with regard to oscillation and response to corticosteroid treatment [38].

The selection of serum markers analyzed in the presented study was dictated by availability in routine diagnostic testing and the parameters represent common laboratory values which are monitored during checkpoint inhibition therapy as well as in the treatment of autoimmune diseases. Especially S100B has proven its usefulness in monitoring metastatic malignant melanoma [33,34]. LDH, a common tumor marker used for many tumors, did not prove helpful in our patients. One reason could be the limited dynamic range of LDH values in the serum of cancer patients which encompasses approximately one order of magnitude whereas S100B in the serum of melanoma patients ranges from below 0.11 µg/L to levels above 10.0 µg/L, suggesting a significantly greater sensitivity. Moreover, the half-life of LDH lies in the range of 100 h whereas the half-life of S100B in the serum is about 2 h [39,40]. Shorter serum availability of a marker molecule is considered an advantage for serial sampling of biomarkers [41].

Nevertheless, S100B as well as IL-6 might not represent the most sensitive and most useful serum or blood markers for the detection of tolerance breakage in melanoma. A disadvantage of S100B as a tumor marker of malignant melanoma is the possible loss of S100B expression during malignant progression which would disrupt the dependency of measured serum S100B levels to the cell loss fraction and immunologic cell death (see Figure 2) [42]. In the future, this limitation may be overcome by measuring as tumor markers for malignant melanoma or for other cancers the amount of tumor mutation-specific cell free DNA in the serum which has a short half-life and is independent of protein expression [43].

A rise of serum IL-6 levels was associated with immune-related adverse events as well as with release of tumor marker in the majority of the treated melanoma patients presented in this study. A study from 2019 analyzed the correlation of different cytokines including IL-6 and irAE during checkpoint inhibitor treatment of various cancers on a two to three weeks basis and identified that the irAE group (*n*  =  16) had significantly elevated levels of five cytokines (IL-6, CXCL2, CCL20, CXCL8 and CCL23) compared to healthy controls [44]. Moreover, the authors stated that the increase in cytokines/chemokines at two–three weeks and at six weeks was significantly greater in the irAE group. Our data is very compatible with their findings underlining the utility of IL-6 monitoring for detection of irAEs under immune checkpoint inhibitor therapy.

Nevertheless, two patients in our study demonstrated an immediate and parallel decrease of IL-6 and S100B serum levels after administration of checkpoint inhibitor treatment which might result from a direct suppressive effect of PD-1 blockade on melanoma cell cycle. Under the assumption that IL-6 can be excreted by melanoma cells and may stimulate autocrine growth [36], the parallel decrease of IL-6 in these patients might suggest that blockade of PD-1 did interrupt this putative autocrine loop. High levels of IL-6 secreted by melanoma metastases might mask the IL-6 protein signature deriving from immune activation. Moreover, progression of melanoma metastases secreting IL-6 should result in a parallel increase of both S100B and IL-6, thus imitating an immunologic response (Figure 5). On the other hand, measuring IL-6 together with a tumor marker under checkpoint inhibitor treatment might help to identify melanomas and other cancers depending on autocrine or paracrine IL-6 signaling [45].

Furthermore, it has to be considered that IL-6 might not be the decisive cytokine triggering an anticancer immune response as anti-IL-6 treatment has been administered in order to treat checkpoint inhibitor-associated adverse effects without overt loss of treatment efficacy [15,46]. In two patients (Figure 7A and Figure 8) the peak of S100B occurred 14 days after the observed peak of IL-6 which might suggest that IL-6 reflects immune activation but may not be implicated in the local anticancer immune reaction within the cancer tissue. Expanding the search for immune markers other than IL-6 which might correlate even more closely to anticancer immune activation and which might not be biased by autocrine or paracrine IL-6 secretion seems mandatory for future research in the field of immune monitoring.

The study was not designed to evaluate sensitivity and specificity of a predictive biomarker based on the identification of the concurrent release in the serum of tumor markers and of markers of immune activation as its aim was to identify a putative molecular serum signature in the first place. The number of analyzed patients is low, and the results must be considered preliminary evidence of an exploratory research. A search for more specific markers of release of cancer derived biomolecules and biomolecules reflecting immune activation as well as a prospective study using more patients under more controlled conditions might be necessary to address this open question.

The presented algorithm which consists of measuring the time course of tumor marker release as well as of a marker of immune activation in the serum of cancer patients in order to identify the time point of parallel increase of both markers which would suggest tolerance breakage under immune checkpoint inhibitor treatment should, in principle, be applicable to all cancer entities. A general requirement would be a high sensitivity with a wide dynamic range in the serum of the chosen markers, a short half-life of the markers in the serum, as well as a blood sampling which is frequent enough to identify serum level oscillations of the markers and to allow area under the curve (AUC) quantification of the released markers. A limitation to the described approach could stem from multi-organ metastases demonstrating mixed response to treatment which would result in opposing trends of tumor marker release masking an existing anticancer immunologic response.

Analyzing the kinetics of immunologic anticancer response during immune checkpoint inhibition in patients might not only provide a predictive biomarker under therapy but it might also be used in the future to optimize treatment efficacy. Advanced AI approaches for data extraction and analysis could be implemented which might make it possible to identify patient-specific molecular signatures of the immune system and of cancer cell cycle regulation [47]. Such molecular signatures could then be used to calibrate treatment and possibly to quantify the optimal intervention timing and dosage to maximize the benefit to each individual patient in immune checkpoint inhibitor therapy.

## 5. Materials and Methods

All patients described in the presented study were adults and were treated for metastasized malignant melanoma at the Euregio Skin Cancer Center of the Department of Dermatology at the University Hospital of the RWTH-Aachen according to German treatment guidelines. Routine blood laboratory values which included interleukin 6 (IL-6), the tumor markers LDH and S100B as well as blood cell count with white cell differentiation were analyzed retrospectively for a concurrent increase in tumor marker release and immune activation. The investigation did not include any intervention in the sense of an interventionary study. In most patients, blood was drawn every week between the first five treatment administrations. Weekly clinical and laboratory controls were routinely performed in order to detect severe and potentially lethal immune-mediated adverse effects at an early time point allowing efficient treatment. Some patients consented to an additional blood sampling one day after checkpoint inhibitor administration. Deviations from the recommended timing of checkpoint inhibitor administration were dictated by the clinical occurrence of adverse effects. All patients provided their written informed consent. This non-interventional study was approved by the local ethics committee of the RWTH-Aachen University (approval code EK153/12, 12 December 2012).

Routine blood samples were processed within one day considering that S100B is sensitive to delayed processing resulting in false positive values. Routine laboratory tests in melanoma patients treated with immune checkpoint inhibitors comprised a blood cell count with white cell differentiation as well as determination of serum levels of c-reactive protein and interleukin-6 as markers of infection and autoimmunity. Serum levels of S100B and lactate dehydrogenase were routinely measured as tumor markers for monitoring malignant melanoma. Serum levels of sodium and potassium were assessed in order to detect hypophysitis. Free triiodothyronine (fT3), free thyroxine (fT4) and thyroid-stimulating hormone (TSH) were quantified for detection of hypophysitis or hyperthyroiditis. Immune-mediated liver damage was assessed by measuring gamma-glutamyl transferase (γ-GT), aspartate aminotransferase (AST), alanine aminotransferase (ALT) and bilirubin. Myositis was evaluated by creatine kinase or myoglobin and renal disease was monitored by creatinine levels.

The search for a concurrent elevation of markers of immune activation and tumor marker release as well as release of organ derived proteins indicating immune-mediated organ damage was performed solely by graphic visualization of time courses of laboratory values during treatment cycles as represented in Figure 1, Figure 2, Figure 3, Figure 4, Figure 5, Figure 6, Figure 7 and Figure 8. Laboratory values were normalized with 100% indicating the upper limit of normal (ULN) for visualization. ULN for displayed laboratory values were 0.6/nL for monocyte count, 0.46/nL for eosinophil count, <5 mg/L for CRP, <7 pg/mL for IL-6, <250 U/L for LDH, <11 μg/L for S100B, <190 U/L for creatine kinase, 72 μg/L for myoglobin and <50 U/L for ALT.

Treatment with immune checkpoint inhibitors was performed according to EU approval. Ipilimumab monotherapy was performed at a dosage of 3 mg/kg every 3 weeks (patients no. 1–5). Nivolumab was administered either at a dosage of 3 mg/kg every 2 weeks (patients no. 1 and 4) or at a fixed dosage of 480 mg every four weeks (patients 8, 13–15 and 17). Pembrolizumab was given either at a dosage of 2 mg/kg every three weeks (patients 5–7, 9 and 11) or at a fixed dosage of 200 mg every three weeks (patient 10). Nivolumab plus ipilimumab combination treatment was performed at a dosage of 1 mg/kg nivolumab and 3 mg/kg ipilimumab every three weeks during the first four cycles (patients no. 12, 14–16). Patient no. 17 was treated with the dosage of 3 mg/kg nivolumab and 1 mg/kg ipilimumab every three weeks approved for renal cell cancer after multidisciplinary tumor board discussion in order to reduce toxicity.

Anonymized clinical and laboratory data are provided in Appendix A.

## Figures and Tables

**Figure 1 ijms-21-02020-f001:**
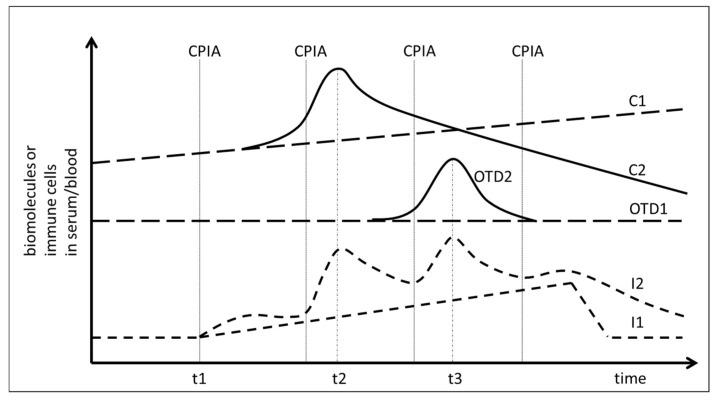
Analysis of temporal correlation of immune activation and release of biomolecules into the serum of cancer patients treated with checkpoint inhibition (CPI). t1: Start of CPI. CPIA: Administration of checkpoint inhibitors. C1: release of cancer-derived proteins or DNA in the absence of a response to CPI. C2: Release of cancer-derived proteins or DNA in response to CPI at time point t2. OTD1: Release of organ tissue derived biomolecules without immune mediated tissue damage. OTD2: Increase of organ tissue derived biomolecules due to treatment-induced and immune-mediated tissue damage at time point t3. I1: Continuous increase of immune activity under CPI. I2: Non-continuous phasic increase of immune activity under CPI.

**Figure 2 ijms-21-02020-f002:**
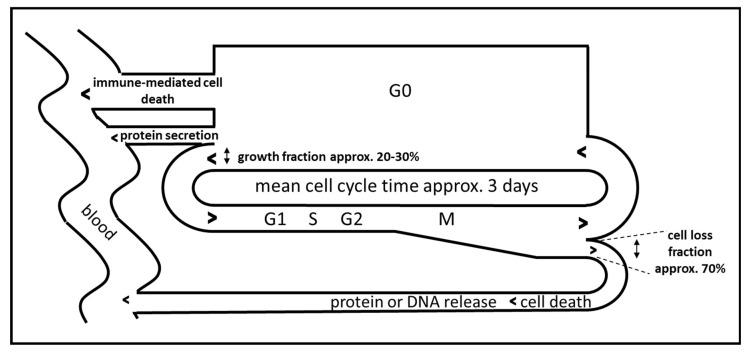
Idealized model of cell cycle-dependent immune-mediated release of biomarkers during immunotherapy of cancer, adapted from [32].

**Figure 3 ijms-21-02020-f003:**
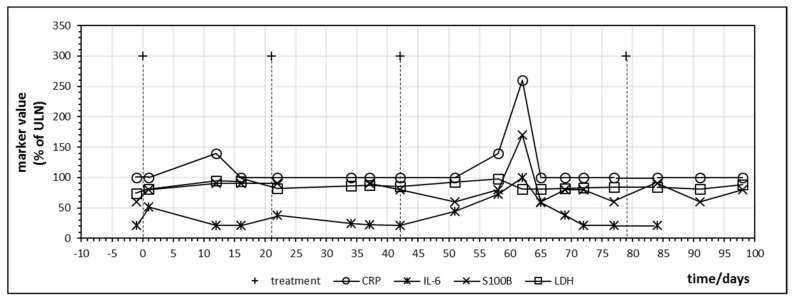
Time course of immune markers c-reactive protein (CRP) and IL-6 and tumor markers S100B and lactate dehydrogenase (LDH) during checkpoint inhibitor treatment starting at day 0. Patient no. 1, four cycles of ipilimumab, delay in the administration of the fourth cycle was due to gastral bleeding.

**Figure 4 ijms-21-02020-f004:**
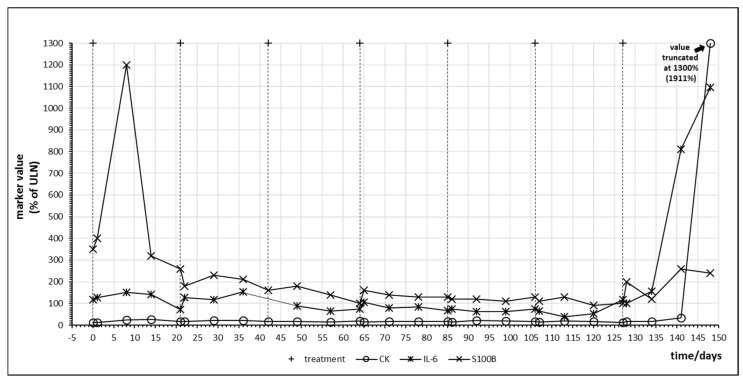
Time course of immune, tumor and cell damage (creatine kinase (CK)) markers during checkpoint inhibitor treatment starting at day 0. Patient no. 5, 7 cycles of pembrolizumab.

**Figure 5 ijms-21-02020-f005:**
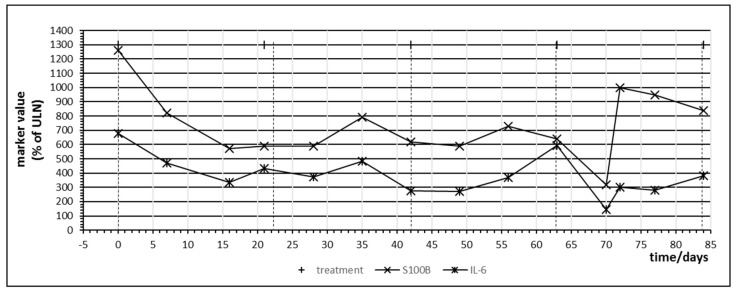
Time course of immune and tumor markers during checkpoint inhibitor treatment starting at day 0. Patient no. 6, first four cycles of pembrolizumab.

**Figure 6 ijms-21-02020-f006:**
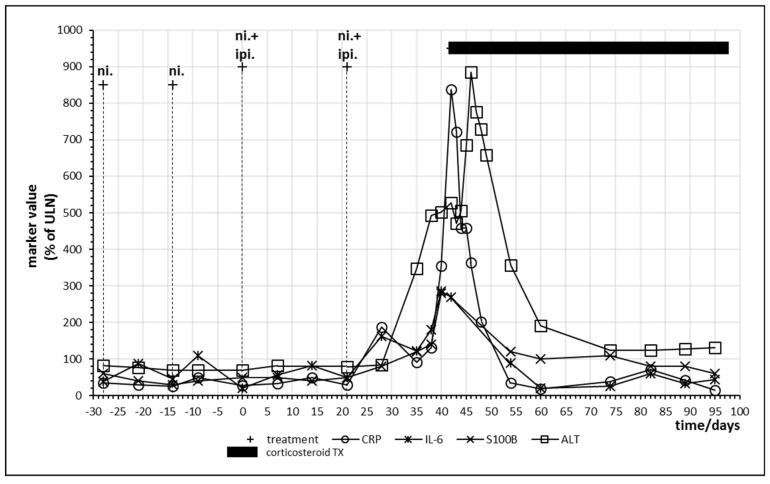
Time course of immune, tumor and cell damage (ALT) markers during checkpoint inhibitor treatment. Patient no. 8, last two cycles of nivolumab; two cycles of nivolumab plus ipilimumab starting at day 0.

**Figure 7 ijms-21-02020-f007:**
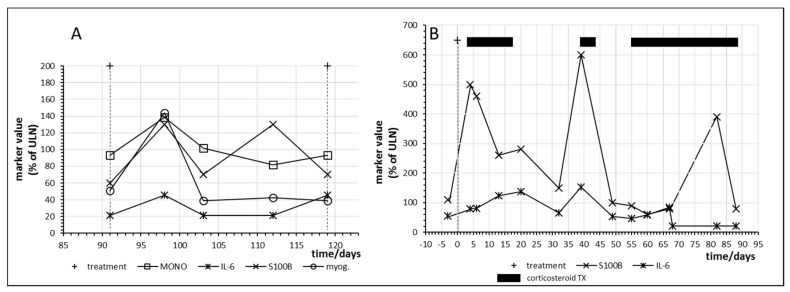
Time course of immune, tumor and cell damage (myoglobin (myog.)) markers during checkpoint inhibitor treatment starting at day 0. (**A**) Patient no. 15, fourth cycle of nivolumab; (**B**) Patient no. 16, first cycle of nivolumab plus ipilimumab combination treatment. MONO = monocytes.

**Figure 8 ijms-21-02020-f008:**
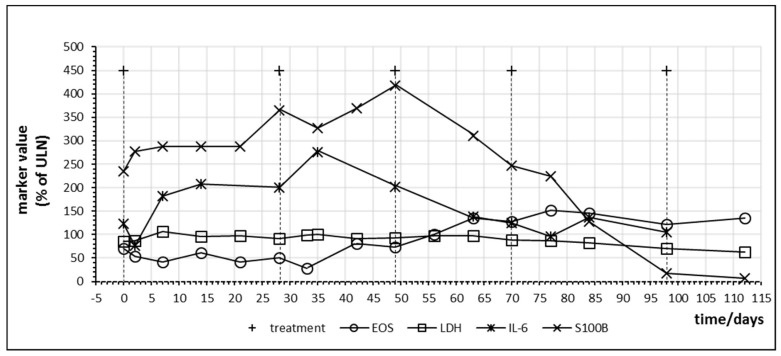
Time course of immune and tumor cell damage markers during checkpoint inhibitor treatment starting at day 0 in patient no. 17 receiving four cycles of nivolumab plus ipilimumab combination treatment followed by nivolumab monotherapy. Interval between the first and second cycle was prolonged due to gastritis of unknown cause. For better visualization, S100B levels are plotted at ¼ of actual values. EOS = eosinophiles.

**Table 1 ijms-21-02020-t001:** Predictive biomarkers sensu stricto of checkpoint inhibitor therapy positively associated with clinical response.

Biomarker	Cancer Entity	Reference
Enhanced programmed cell death ligand 1 (PD-L1) expression in the tumor	melanoma, non-small cell lung cancer (NSCLC), renal-cell carcinoma, prostate cancer, colorectal cancer	[5,6,7,8]
Presence of CD8+tumor-infiltrating lymphocytes	melanoma, NSCLC, renal-cell carcinoma, colorectal cancer	[5,6,7,9,10]
High tumor mutational burden or neoantigen burden	melanoma, NSCLC, colorectal cancer, urothelial carcinoma	[5,11,12]
Presence of intratumoral major histocompatibility complex (MHC) class II expression	melanoma	[5,13]
Presence of intratumoral interferon-γ-immune gene signature	melanoma, head and neck cancer	[5,14]
Low interleukin (IL)-6 expression in the tumor	colorectal cancer	[15]
Peripheral blood count: low absolute neutrophils, low absolute monocytes, low myeloid-derived suppressor cells, high FoxP3+ regulatory T cells, high lymphocytes, high eosinophils, high CD19−HLA-DR+ myeloid cells, high CD14+CD16b−HLA-DRhi monocytes	melanoma, NSCLC	[5,16,17,18]
Low level of c-reactive protein (CRP) in the serum, low relative eosinophil count	uveal melanoma	[19]
Serum proteome analysis: BDX008	melanoma	[20]

**Table 2 ijms-21-02020-t002:** Treatment response monitoring (predictive) biomarkers of checkpoint inhibitor therapy.

Biomarker	Cancer Entity	Reference
Peripheral blood count: decreasing FoxP3+ regulatory T cells, increasing absolute lymphocytes, increasing eosinophils, decrease of the neutrophil-to-lymphocyte ratio, decrease of HLA-DR monocytes, increase of total dendritic cells	melanoma, NSCLC	[18,21,22]

**Table 3 ijms-21-02020-t003:** Biomarkers for monitoring immune activation in inflammatory and autoimmune diseases.

Biomarker	Autoimmune Disease	Reference
Serum levels: elevated serum amyloid A, Interleukin (IL)-6, IL-8, eotaxin-1	Inflammatory Bowel Disease	[25]
Peripheral blood count: elevated monocytes	Painless autoimmune thyroiditis	[26]
Peripheral blood count: elevated eosinophils	Grave’s disease	[26]
Serum levels: IL-6, IL-8, IL-17, IL-21, tumor necrosis factor (TNF)-α	Autoimmune hepatitis	[27,28]
Peripheral blood count: elevated eosinophils	Autoimmune pneumonitis	[29]

**Table 4 ijms-21-02020-t004:** Tumor marker release, immunologic adverse effects and response to therapy in 14 patients treated with immune checkpoint inhibition.

no.	TNM at TX	TX (Cycles)	TMR IM	TMR Day	irAE (IM)	irAE. Day	best resp.	Survival from TX, Months
1	pT3a, N3c, M1a	ipi. (4)	IL-6	62	-		PR	>89
		nivo. (11)	-		thyroiditis (mono.)	28	PR	
2	pT1a, N3, M1c	ipi. (4)	IL-6	81	myositis	81	PD	8.5
					hepatitis			
					(IL-6, CRP, mono.)			
3	pT4, N1b, M1c	ipi. (3)	-	-	thyroiditis	49	PD	19
4	pT3a, N3, M1c	ipi. (4)	IL-6	16	hepatitis	77	PR	14
					(mono., eos.)			
		nivo. (7)	paradox		-	-		
5	pT3a, N3, M1c	ipi. (3)	IL-6	1	colitis	70	PR	>6
					(eos.)			
		pemb. (33)	IL-6	8, 141	myositis (IL-6)	148	CR	
6	pTx, N3, M1c	pemb. (7)	paradox	-	-	-	PD	8
7	pT3b, N3, M1c	pemb. (8)	IL-6	5	pneumonitis	188	PR	>11
			mono.		(IL-6)			
8	pT2a, N2b, M1c	nivo. (18)	-	-	-	-	PR	>51
		nivo.+						
		ipi. ((2), 11)	IL-6	40	hepatitis (IL-6, CRP)	40	CR	
9	pT4a, N3, M0	pemb. (9)	-	-	-	-	PD	8
10	pT4a, N1a	pemb. (19)	IL-6	71	-	-	PD	>13
11	pTx, N0, M1c	pemb. (28)	IL-6	(21) 141	-	-	CR	>41
12	pT3b, N1b, M1d	nivo.+	IL-6	6	neuritis	6	CR	27
		ipi. (1)						
13	pT4, N1c, M0	nivo. (3)	-	-	-	-	PD	14
14	pTx, Nx, M1b	nivo.+	IL-6	28	thyroiditis, hepatitis	35	CR	>25
		ipi. (1)			(IL-6, CRP)			
		nivo. (7)	-		Pneumonitis (IL-6, CRP, mono.)	182	CR	
15	pT2b, N3c, M1c	nivo.+	IL-6	7 (112) *	thyroiditis,	21	CR	>35
		ipi. (2)			(IL-6, mono.)			
		nivo. (8)	IL-6		Myositis (IL-6)	98	CR	
16	pTx, N2, M1a	nivo.+	IL-6	(4) 39	rash	4	CR	>11
		ipi. (1)			hypophysitis, pneumonitis	20		
					(IL-6, mono.)	20		
17	pT3a, N1b, M1c	nivo. (1)	IL-6 *	28 *	vitiligo	70	PR	>8
		nivo. + ipi ((4), 10)	CRP		(eos.)			

TNM: AJCC 2017 TNM classification of malignant tumors; TX: checkpoint inhibitor treatment, TMR: tumor marker response, IM: immune marker associated with response or immune-related adverse events. ipi. = ipilimumab; nivo. = nivolumab; pemb. = pembrolizumab; mono. = monocytes; eos. = eosinophiles. * S100B increase 14 days after IL-6 peak.

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
