# Peer review of "Correlative Monitoring of Immune Activation and Tissue Damage in Malignant Melanoma—An Algorithm for Identification of Tolerance Breakage During Immune Checkpoint Inhibitor Therapy of Cancer"

_ijms, 2020, doi:10.3390/ijms21062020_

Round 1
Reviewer 1 Report
The issue is very interesting; however, in my opinion authors cannot present their paper as an introduction of an "universal algorithm". In fact, the paper is substantially a case-report study, as it only reports some interesting observations from 14 advanced melanoma cases. A statistical analysis taking into account data on objective response and mostly patients survival would be necessary to support the presented data.
Author Response
Reviewer's comments:
The issue is very interesting; however, in my opinion authors cannot present their paper as an introduction of an "universal algorithm". In fact, the paper is substantially a case-report study, as it only reports some interesting observations from 14 advanced melanoma cases. A statistical analysis taking into account data on objective response and mostly patients survival would be necessary to support the presented data.
Author's reply:
We fully agree to the criticism that presenting our limited observational data as an universal algorithm is quite exaggerated. We have therefore modified the title to “Correlative monitoring of immune activation and tissue damage in malignant melanoma – An algorithm for identification of tolerance breakage during immune checkpoint inhibitor therapy of cancer” by introducing “melanoma” and omitting “universal”.
We are aware that even this softened title might still appear disproportionate. We have identified additional three patients with a dense coverage of laboratory values under immune checkpoint inhibitor therapy and included the patients in order to strengthen our claims.
On the other hand, it should be considered that we present a unique data set. To our knowledge no data have been published up to now which have analyzed the effects of immune checkpoint inhibition treatment on laboratory values on a weekly basis and the graphs that we present demonstrate that in most cases the observed rise in S100B as well as of IL-6 lasts only a few days, which means, that it may not be observed when determining blood values every two to four weeks which is the normally used frequency in oncology. Moreover, S100B is a melanoma tumor marker which is mainly used in Europe and our data presume that LDH is not sensitive enough and has a half-life which reduces its capability to detect small variations in tumor marker release. IL-6 has, to our knowledge, not been used on a regular basis in order to detect immune related side effects. Nonetheless, we have found a recent article from 2019 which analyzed CRP every day to every second day in a case of checkpoint inhibitor associated neutropenia as well as IL-6 before and during the immune related adverse event (irAE) and the published time courses closely resemble our figures [Naqash et al. 2019]. Another study from 2019 has analyzed the correlation of different cytokines including IL-6 and irAE during checkpoint inhibitor treatment of various cancers on a two to three weeks basis and identified that the irAE group (n = 16) had significantly elevated levels of five cytokines (IL-6, CXCL2, CCL20, CXCL8 and CCL23) compared to healthy controls [Khan et al. 2019]. Moreover, they stated that “conversely, the fold increase in cytokines/chemokines at 2–3 weeks and at 6 weeks (particularly for CXCL9 and CXCL10) was significantly greater in the irAE group”. Our data is very compatible with their findings underlining the utility of IL-6 monitoring for detection of irAEs under immune checkpoint inhibitor therapy of other cancers than malignant melanoma. We have included both references in our revised manuscript. The emerging evidence from these two very recent studies concerning the parallel increase of IL-6 and immune activation during irAE under immune checkpoint inhibitor treatment lends support to our hypothesis that a parallel increase of serum IL-6 and S100B might be a molecular marker of immune activation against the treated cancer. We therefore believe that the described approach, possibly with an improved set of biomarkers could indeed represent a universal approach to better monitor efficacy and side effects of immune checkpoint inhibitor treatment.
With regards to the comment “A statistical analysis taking into account data on objective response and mostly patients survival would be necessary to support the presented data.” we may only state in the manuscript that “The study was not designed to evaluate sensitivity and specificity of a predictive biomarker based on the identification of the concurrent release in the serum of tumor markers and of markers of immune activation as its aim was to identify a putative molecular serum signature in the first place. The number of analyzed patients is low, and the results must be considered preliminary evidence of an exploratory research.”
We, nevertheless, included overall survival data in table 4 as the question whether demonstration of the IL-6/S100B serum signature is associated with overall survival is a very justified request. Our own position is that the analysis of IL-6/S100B and tissue damage markers on a weekly basis provide valuable information on the effects of checkpoint inhibitor treatment in individual patients. Nonetheless, multiple factors influence the values and their significance which complicate interpretation, especially as multiple laboratory values will be generated over treatment time. For example, concurrent infections will enhance IL-6 significantly. Patients with a limited tumor mass will have low to undetectable S100 values but will respond best to treatment. Therefore, we suggested to search for more sensitive tumor marks. Likewise, patients with a high tumor burden often have an already strongly activated immune system with high levels of IL-6 either from the immune system or from auto/paracrine IL-6 secretion. In some cases, we might observe the “paradox” reaction of concomitant reduction of IL-6 and S100B which might mask a parallel immune activation by immune checkpoint inhibition. We assume that an AI-based approach is needed to disentangle the different influencing factors and that a mere statistical analysis might not be able to answer the question.
We hope that our reply to the reviewer’s comments and our revised manuscript may be accepted and we would like to thank the reviewer for his comments as it enabled us to improve several weak points in our manuscript. We have included three additional patients and one additional figure and have amended the Discussion, the Material and Methods as well as the References section.
Yours sincerely
Albert Rübben
References:
Naqash AR, Appah E, Yang LV, Muzaffar M, Marie MA, Mccallen JD, Macherla S, Liles D, Walker PR. Isolated neutropenia as a rare but serious adverse event secondary to immune checkpoint inhibition. J Immunother Cancer. 2019 Jul 5;7(1):169.
Khan S, Khan SA, Luo X, Fattah FJ, Saltarski J, Gloria-McCutchen Y, Lu R, Xie Y, Li Q, Wakeland E, Gerber DE. Immune dysregulation in cancer patients developing immune-related adverse events. Br J Cancer. 2019 Jan;120(1):63-68.
Reviewer 2 Report
- Overall the study looks basic
- The sample size is low
- I would recommend the authors to elaborate the importance of checking IL-6 though there are several pro-inflammatory and inflammatory cytokine available
- More methods needs to be added
Author Response
Reviewer's comment:
1) Overall the study looks basic
Author's reply:
We agree that the study is basically observational and that the conclusions stem from preliminary evidence of an exploratory research. We have therefore modified the title to “Correlative monitoring of immune activation and tissue damage in malignant melanoma – An algorithm for identification of tolerance breakage during immune checkpoint inhibitor therapy of cancer” by introducing “melanoma” and omitting “universal” and we have included the statement “The number of analyzed patients is low, and the results must be considered preliminary evidence of an exploratory research.”.
On the other hand, it should be considered that we present a unique data set. To our knowledge no data have been published up to now which have analyzed the effects of immune checkpoint inhibition treatment on laboratory values on a weekly basis and the graphs that we present demonstrate that in most cases the observed rise in S100B as well as of IL-6 last only a few days, which means, that it may not be observed when determining blood values every two to four weeks which is the normally used frequency in oncology. Moreover, S100B is a melanoma tumor marker which is mainly used in Europe and our data presume that LDH is not sensitive enough and has a half-life which reduces its capability to detect small variations in tumor marker release. IL-6 has, to our knowledge, not been used on a regular basis in order to detect immune related side effects with the exception of two recent studies (see answer to comment no. 3) [Naqash et al. 2019, Khan et al. 2019]. No published study provide data on both values on a weekly basis, but given the last two publications and the important implications for immune checkpoint inhibitor therapy, we expect that in the near future more groups will focus on a similar approach as described in our manuscript.
Reviewer’s comment:
2) The sample size is low
Author’s reply:
We agree to the criticism and we have included three additional patients with a dense coverage of laboratory values under immune checkpoint inhibitor therapy. An additional figure has been included in the manuscript.
Reviewer’s comment:
3) I would recommend the authors to elaborate the importance of checking IL-6 though there are several pro-inflammatory and inflammatory cytokine available
Author’s reply:
We are very thankful for this advice. As we have not archived serum samples of our treated patients, we may not check for other cytokines in the patients described in the manuscript. Nonetheless, we have found a recent article from 2019 which analyzed CRP every day to every second day in a case of checkpoint inhibitor associated neutropenia as well as IL-6 before and during the immune related adverse event (irAE) and the published time courses closely resemble our figures [Naqash et al. 2019]. Another study from 2019 has analyzed the correlation of different cytokines including IL-6 and irAE during checkpoint inhibitor treatment of various cancers on a two to three weeks basis and identified that the irAE group (n = 16) had significantly elevated levels of five cytokines (IL-6, CXCL2, CCL20, CXCL8 and CCL23) compared to healthy controls [Khan et al. 2019]. Moreover, they stated that “conversely, the fold increase in cytokines/chemokines at 2–3 weeks and at 6 weeks (particularly for CXCL9 and CXCL10) was significantly greater in the irAE group”. Our data is very compatible with their findings underlining the utility of IL-6 monitoring for detection of irAEs under immune checkpoint inhibitor therapy of other cancers than malignant melanoma. We have included both references in our revised manuscript. The emerging evidence from these two very recent studies concerning the parallel increase of IL-6 and immune activation during irAE under immune checkpoint inhibitor treatment lends support to our hypothesis that a parallel increase of serum IL-6 and S100B might be a molecular marker of immune activation against the treated cancer. We therefore believe that the described approach, possibly with an improved set of biomarkers could indeed represent a universal approach to better monitor efficacy and side effects of immune checkpoint inhibitor treatment.
Reviewer’s comment:
4) More methods needs to be added.
Author’s reply:
We agree that the Methods section is quite short. We have amended the Materials and Methods section with information on tested laboratory values as well as on treatment of the patients.
We hope that our reply to the reviewer’s comments and our revised manuscript may be accepted and we would like to thank the reviewer for his comments as it enabled us to improve several weak points in our manuscript. We have included three additional patients and one additional figure and have amended the Discussion, the Material and Methods as well as the References section.
Yours sincerely
Albert Rübben
References:
Naqash AR, Appah E, Yang LV, Muzaffar M, Marie MA, Mccallen JD, Macherla S, Liles D, Walker PR. Isolated neutropenia as a rare but serious adverse event secondary to immune checkpoint inhibition. J Immunother Cancer. 2019 Jul 5;7(1):169.
Khan S, Khan SA, Luo X, Fattah FJ, Saltarski J, Gloria-McCutchen Y, Lu R, Xie Y, Li Q, Wakeland E, Gerber DE. Immune dysregulation in cancer patients developing immune-related adverse events. Br J Cancer. 2019 Jan;120(1):63-68.
Round 2
Reviewer 1 Report
Authors have modified their paper according my suggestion and completely clarified some unclear points.
Reviewer 2 Report
No specific comments